# calibmsm: An *R* package for calibration plots of the transition probabilities in a multistate model

Alexander Pate[ID][1]*, Matthew Sperrin[1], Richard D. Riley[2,3], Ben Van Calster[4,5], Glen P. Martin[1]

1 Division of Imaging, Informatics and Data Science, University of Manchester, Manchester, United Kingdom, 2 Department of Applied Health Sciences, School of Health Sciences, College of Medicine and Health, University of Birmingham, Birmingham, United Kingdom, 3 National Institute for Health and Care Research (NIHR) Birmingham Biomedical Research Centre, United Kingdom, 4 Department of Development and Regeneration, KU Leuven, Leuven, Belgium, 5 Leuven Unit for Health Technology Assessment Research (LUHTAR), KU Leuven, Leuven, Belgium

* alexander.pate@manchester.ac.uk

## Abstract

### Background and objective

Multistate models, which allow the prediction of complex multistate survival processes such as multimorbidity, or recovery, relapse and death following treatment for cancer, are being used for clinical prediction. It is paramount to evaluate the calibration (as well as other metrics) of a risk prediction model before implementation of the model. While there are a number of software applications available for developing multistate models, currently no software exists to aid in assessing the calibration of a multistate model, and as a result evaluation of model performance is uncommon. ***calibmsm*** has been developed to fill this gap.

### Methods

Assessing the calibration of predicted transition probabilities between any two states is made possible through three approaches. The first two utilise calibration techniques for binary and multinomial logistic regression models in combination with inverse probability of censoring weights, whereas the third utilises pseudo-values. All methods are implemented in conjunction with landmarking to allow calibration assessment of predictions made at any time beyond the start of follow up. This study focuses on calibration curves, but the methodological framework also allows estimation of calibration slopes and intercepts.

### Results

This article serves as a guide on how to use ***calibmsm*** to assess the calibration of any multistate model, via a comprehensive example evaluating a model developed to predict recovery, adverse events, relapse and survival in patients with blood cancer

**Data availability statement:** The package developed in this study is available from the Comprehensive R Archive Network at https://CRAN.R-project.org/package=calibmsm, or can be installed from the following GitHub repository: https://github.com/alexpate30/calibmsm The data for this worked example is provided with the package, and can also be found in the GitHub repository: https://github.com/alexpate30/calibmsm/tree/master/data. This data has also been uploaded to a stable public repository figshare: https://figshare.com/articles/dataset/Data_required_for_running_vignettes_with_calibmsm_R_package/27635844. This data can be used to replicate study findings. The code for derivation of this data is provided in the file: https://github.com/alexpate30/calibmsm/blob/master/data-raw/prepare_pkg_data.R Supplementary material for this work is provided in the form of articles on the GitHub created website: https://alexpate30.github.io/calibmsm/.

**Funding:** This work was supported by funding from the MRC-NIHR Methodology Research Programme [grant number: MR/T025085/1].

**Competing interests:** The authors declare no potential conflict of interest.

after a transplantation. The calibration plots indicate that predictions of relapse made at the time of transplant are poorly calibrated, however predictions of death are well calibrated. The calibration of all predictions made at 100 days post transplant appear to be poor, although a larger validation sample is required to make stronger conclusions.

## Conclusions

*calibmsm* is an R package which allows users to assess the calibration of predicted transition probabilities from a multistate model. Evaluation of model performance is a key step in the pathway to model implementation, yet evaluation of the performance of predictions from multistate models is not common. We hope availability of software will help model developers evaluate the calibration of models being developed.

## 1. Introduction

Risk prediction models enable the prediction of clinical events in either diagnostic or prognostic settings [1] and are used widely to inform clinical practice. A multistate model [2] may be used when there are multiple outcomes of interest, or when a single outcome of interest may be reached via intermediate states. For example, prediction of death after local recurrence or distant metastasis in patients with breast cancer following surgery [3]; prediction of death following progression of chronic kidney disease [4]; prediction of non-AIDS events and death in individuals living with HIV [5]. Using a multistate model for prediction is important when the development of an intermediate condition occurring post index date may have an impact on the risk of future outcomes of interest. Risk prediction models developed for use in clinical practice should be evaluated in a relevant cohort, or preferably multiple settings/cohorts, prior to implementation [6]. If the intended use of this model is known, targeted validation in a specific setting may be preferred [7]. A key part of the validation process is assessment of the calibration of the model [8]. Calibration assesses whether the predicted risks match the observed event rates in the cohort of interest. Ideally calibration curves should be produced to estimate observed event rates as a function the predicted risks over the entire distribution of predicted risk. This corresponds to a moderate assessment of calibration [9]. Methodology on this topic is well developed for binary outcomes [9], survival outcomes [10,11], and survival outcomes in the presence of competing risks [12,13], however less so for multistate models, where there is often interest in prediction of more than one outcome state, and in predictions made at landmark times.

The *R* [14] package *mstate* provides a comprehensive set of tools to develop a multistate model and estimate patient-specific predictions for a continuously observed multistate survival process [15]. *mstate* focuses on non-parametric and semi-parametric multistate models where the cause-specific hazards have been fitted using cox-proportional hazards models. The *flexsurv* package [16] builds on the functionality of *mstate*, allowing users to fit parametric multistate models (still using

the cause-specific hazards approach), as well as an approach that uses mixture models. Both *mstate* and *flexsurv* allow fitting of clock-forward (Markov) and clock reset (Semi-Markov) models. The *SemiMarkov* package [17] contains functions specifically for fitting semi-Markov models. The *msm* package [18] focuses on fitting multistate models to continuous time processes that are observed at arbitrary times (panel data). The *flexmsm* package [19] provides a general estimation framework for multistate Markov processes, with flexible specification of the transition intensities. Transition intensities can be specified through Generalised Additive Models, and allows models with forward and backward transitions to be fitted. The Lexis functions from the *Epi* package [20] provide a way to represent and manipulate data from multistate models, and provides an interface to the *mstate*. For a full list of packages available for fitting multistate models, see https://cran.r-project.org/web/views/Survival.html.

There are a number of *R* packages which enable the estimation of calibration curves for models predicting continuous, binary, or survival outcomes [21–23]. However, despite a wide range of packages for developing multistate models, currently no software exists to aid researchers in assessing the calibration of a multistate model that has been developed for the purposes of individual risk prediction. The *R* package *calibmsm* has been developed to enable researchers to estimate calibration curves and scatter plots for the transition probabilities between any two states. The methodology is based on three approaches outlined previously [24], which focused on assessing the calibration of the transition probabilities out of the starting state. The work in this paper extends the framework to assess the calibration of transition probabilities out of any state *j* at any time *s* using landmarking [25,26], provides more details on estimation of the inverse-probability of censoring weights (where relevant), and demonstrates the process for estimating confidence intervals. *calibmsm* is available from the Comprehensive R Archive Network at https://CRAN.R-project.org/package=calibmsm, or can be installed from the following GitHub repository [27].

```
# install.packages("calibmsm")
# devtools::install_github("alexpate30/calibmsm")
```

De Wreede et al. [15], used data from the European Society for Blood and Marrow Transplantation [28] to showcase how to develop a multistate model for clinical prediction of outcomes after bone morrow transplantation in leukemia patients (Fig 1). In this study, we show how to assess the calibration of a model developed on the same EBMT data as a way of illustrating the syntax and workflows of *calibmsm*. This clinical example also highlights some important differences between the methods in how they deal with informative censoring and computational feasibility, which may impact future

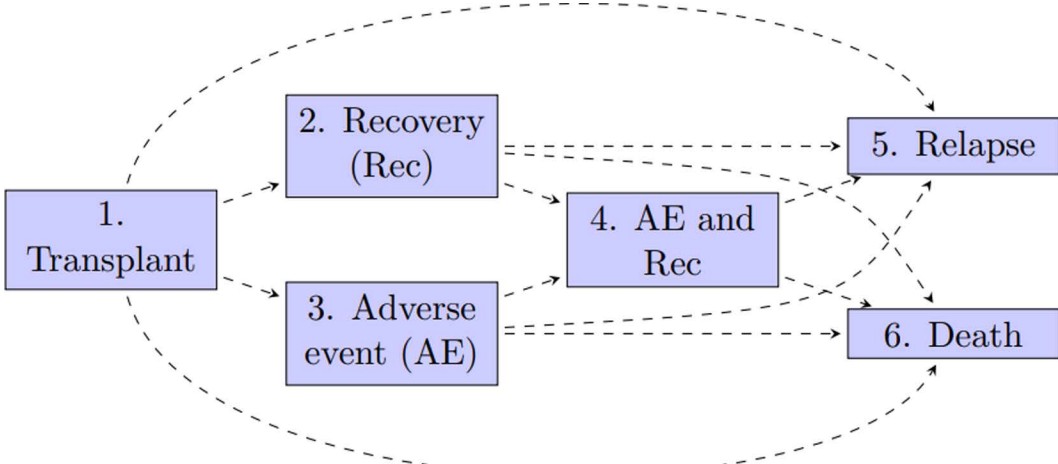

**Fig 1. A six-state model for leukemia patients after bone marrow transplantation.** Figure taken from De Wreede et al [15].

uptake of the methods. Details on the methodology are given in section 2. The clinical example, including steps for data preparation and production of calibration plots are given in section 3. Section 4 contains a discussion and summary.

## 2. Methods and theory

### 2.1 Setup

Let $X(t) \in \{1, \ldots, K\}$ be a multistate survival process with $K$ states. We assume a multistate model has already been developed and we want to assess the calibration of the predicted transition probabilities, $\hat{p}_{j,k}(s,t)$, in a cohort of interest. The transition probabilities are the probability of being in state $k$ at time $t$, if in state $j$ at time $s$, where $s < t$. To assess the calibration of the multistate model, we must estimate observed event probabilities:

$$o_{j,k}(s,t) = P[X(t) = k | X(s) = j, \hat{p}_{j,k}(s,t)].$$

In a well calibrated model, the transition probabilities will be equal to the observed event probabilities.

In the absence of censoring, $o_{j,k}(s,t)$ can be estimated using cross sectional calibration techniques in a landmark [25,26] cohort of individuals who are in state $j$ at time $s$ (i.e., methods to assess the calibration of models predicting binary or multinomial outcomes). In the presence of censoring, calibration must be assessed in this landmark cohort of individuals either using cross sectional techniques in combination with inverse probability of censoring weights, or through pseudo-values. These approaches were previously proposed and evaluated in a simulation study [24], but were restricted to assessing calibration out of the starting state at time $s = 0$. The theory is summarised and revised to allow assessment of calibration out of any state $j$ at any time $s$ in sections 2.2–2.6.

### 2.2 Binary logistic regression with inverse probability of censoring weights (BLR-IPCW) calibration curves

The first approach produces calibration curves using a framework for binary logistic regression models in conjunction with inverse probability of censoring weights to account for informative censoring (BLR-IPCW). Let $I_k(t)$ be an indicator for whether an individual is in state $k$ at time $t$. $I_k(t)$ is then modeled using a flexible approach with $\hat{p}_{j,k}(s,t)$ as the sole predictor. This model is fit in the landmark cohort (in state $j$ at time $s$) of individuals who are also still uncensored at time $t$. This cohort is weighted using inverse probability of censoring weights (see section 2.4). We suggest using a loess smoother [29]:

$$I_k(t) = loess(\hat{p}_{j,k}(s,t)) \tag{1}$$

or a logistic regression model with restricted cubic splines [30]:

$$logit(I_k(t)) = rcs(logit(\hat{p}_{j,k}(s,t))) \tag{2}$$

Any flexible model for binary outcomes could be used, but these are the most common and are implemented in this package. Observed event probabilities $\hat{o}_{j,k}(s,t)$ are then estimated as fitted values from these models. These are commonly referred to as 'predicted-observed risk' values. The calibration curve is plotted using the set of points $\{\hat{p}_{j,k}(s,t), \hat{o}_{j,k}(s,t)\}$. To obtain unbiased calibration curves, the assumption that each outcome $I_k(t)$ is independent from the censoring mechanism in the reweighted population must hold.

### 2.3 Multinomial logistic regression with inverse probability of censoring weights (MLR-IPCW) calibration scatter plots

The second approach produces calibration scatter plots using a framework for multinomial logistic regression models with inverse probability of censoring weights (MLR-IPCW). Let $I_X(t)$ be a multinomial indicator variable taking values $I_X(t) \in \{1, \ldots, K\}$ such that $I_X(t) = k$ if an individual is in state $k$ at time $t$. The nominal recalibration framework

of Van Hoorde et al. [31,32], is then applied in the landmark cohort of individuals uncensored at time $t$, weighted using inverse probability of censoring weights (section 2.4). First calculate the log-ratios of the predicted transition probabilities:

$$\widehat{LP}_k = \ln\left(\frac{\hat{p}_{j,k}(s,t)}{\hat{p}_{j,k_{ref}}(s,t)}\right)$$

Then fit the following multinomial logistic regression model:

$$\ln\left(\frac{P[I_X(t) = k]}{P[I_X(t) = k_{ref}]}\right) = \alpha_k + \sum_{h=2}^{K} \beta_{k,h} * sm_k\left(\widehat{LP}_h\right)$$

where $k_{ref}$ is an arbitrary reference category which can be reached from state $j$, $k \neq k_{ref}$ takes values in the set of states that can be reached from state $j$, and where $sm$ is a vector spline smoother [33]. Observed event probabilities $\hat{o}_{j,k}(s,t)$ are then estimated as fitted values from this model. This results in a calibration scatter plot rather than a curve due to all states being modeled simultaneously, as opposed to BLR-IPCW, which is a "one vs all" approach. The scatter occurs because the observed event probabilities for state $k$ vary depending on the predicted transition probabilities of the other states. This is a stronger [9] form of calibration than that evaluated by BLR-IPCW, and will also result in observed event probabilities which sum to 1. In future iterations of **calibmsm** functionality will be added to produce smoothed curves estimated from these scatter plots. To obtain unbiased calibration curves, the assumption that the outcome $I_X(t)$ is independent from the censoring mechanism in the reweighted population must hold.

## 2.4 Estimation of the inverse probability of censoring weights

The estimand for the weights is $w_j(s,t)$, the inverse of the probability of being uncensored at time $t$ if in state $j$ at time $s$:

$$w_j(s,t) = \frac{1}{P[t_{cens} > t | t > s, \; X(s) = j, \; \mathbf{Z}, \; \mathbf{X}(t)]} \tag{3}$$

where $\mathbf{X}(t)$ denotes the history of the multistate survival process up to time $t$, including the transition times, and $\mathbf{Z}$ is a set of baseline predictor variables believed to be predictive of the censoring mechanism. Note that $\mathbf{Z}$ may be the same as, but is not restricted to, the variables used for prediction when developing the multistate model. First the estimator $\hat{P}[t_{cens} > t | t > s, \; X(s) = j, \; \mathbf{Z}]$ is calculated by developing an appropriate survival model. The outcome in this model is the time until censoring occurs. Moving into an absorbing state prevents censoring from happening and is treated as a censoring mechanism in this model (i.e., a competing risks approach is not taken when fitting this model). $\mathbf{X}(t)$ is explicitly conditioned on when defining $w_j(s,t)$ because the weights must reflect that censoring can no longer be observed for an individual if they enter an absorbing state at some time $s < t_{abs} < t$. Therefore, in **calibmsm**, the weights are estimated as:

$$\hat{P}[t_{cens} > t | t > s, \; X(s) = j, \; \mathbf{Z}, \mathbf{X}(t)] = \hat{P}[t_{cens} > \min\{t, t_{abs}\} | t > s, \; X(s) = j, \; \mathbf{Z}]$$

Note that if the censoring mechanism is not conditionally independent from the outcome process $\mathbf{X}(t)$ given $\mathbf{Z}$, i.e., the rate of censoring changes depending on outcome state occupancy, then this approach will be invalid. Instead, the outcome history up until time $t$ must be conditioned on when estimating the weights, as specified in equation (3). By default, $\hat{P}[t_{cens} > t | t > s, \; X(s) = j, \; \mathbf{Z}]$ is estimated in **calibmsm** using a cox proportional hazards model where all predictors $\mathbf{Z}$ are assumed to have a linear effect on the log-hazard. These conditions are highly restrictive.

Users can therefore also input their own vector of weights which is strongly recommended. *calibmsm* has been developed to provide a framework for assessing calibration, and the packages' primary purpose is not estimation of inverse probability of censoring weights, however, future versions may allow for more flexible models to estimate the weights. Given the BLR-IPCW and MLR-IPCW approaches are both reliant on correct estimation of the weights, we encourage users to take the time to carefully estimate these using a well specified model. The limitations of using the *calibmsm* internal functions for estimating the weights in this clinical example (section 4) are discussed in more detail later, and explored in vignette Sensitivity-analysis-for-IPCWs [27].

Stabilised weights can be estimated by multiplying by the weights $w_j(s, t)$ by the mean probability of being uncensored:

$$w_j^{stab}(s, t) = \frac{P[t_{cens} > t | t > s, \ X(s) = j]}{P[t_{cens} > t | t > s, \ X(s) = j, \ \mathbf{Z}, \ \mathbf{X}(t)]}$$

The numerator can be estimated using an intercept only model, and note there is no dependence on $\mathbf{X}(t)$.

Another option is to estimate $w(s, t)$, which is the inverse of the probability of being uncensored at time $t$ if uncensored at time $s$:

$$w(s, t) = \frac{1}{P[t_{cens} > t | t > s, \ \mathbf{Z}, \ \mathbf{X}(t)]}$$

This can be estimated using the same approach as for $w_j(s, t)$, except there is no requirement to be in state $j$ when landmarking at time $s$. If the censoring mechanism is conditionally independent from $X(s)$ after conditioning on $\mathbf{Z}$, then $w(s, t) = w_j(s, t)$, and any consistent estimator for $w(s, t)$ will be a consistent estimator of $w_j(s, t)$. The advantage is that $w(s, t)$ is calculated by developing a model in the cohort of individuals uncensored at time $s$, which is a larger cohort than those uncensored and in state $j$ at time $s$. Therefore $w(s, t)$ will be a more precise estimator than $w_j(s, t)$. On the contrary, if the rate of differs depending on $X(s)$, there is a risk of bias in estimation of the weights. We therefore recommend using the estimator $w_j(s, t)$ unless sample size (number of individuals in state $j$ at time $s$) is low, which may be assessed using sample size formula for prediction models with time-to-event outcomes [34]. If the sample size is deemed insufficient, one may consider using $w(s, t)$, but the risk of bias associated with this estimator must be carefully considered.

We note the importance of using inverse probability of censoring weights, even if the censoring mechanism is believed to be completely at random. When a multistate model has an absorbing state (which is the case for most), entry into this state will prevent censoring from happening. This induces a dependence between the outcome and the censoring mechanism which must be adjusted for using inverse probability of censoring weights. This issue was highlighted in the supplementary material of previous work [24].

## 2.5 Pseudo-value calibration plots

The third approach produces calibration curves using pseudo-values [35,36]. Pseudo-values can be used in place of the outcome of interest in a regression model if some outcomes are not observed due to right censoring. This is the case in models (1) and (2). For certain estimators $\hat{\theta}$ (where $\theta$ estimates the expectation of the outcome it is replacing), the pseudo-value for individual $i$ is defined as:

$$\hat{\theta}^i = n * \hat{\theta} - (n-1) * \hat{\theta}^{-i},$$

where $\hat{\theta}^{-i}$ is equal to $\hat{\theta}$ estimated in a cohort without individual $i$. One such estimator for the outcomes in models (1) and (2) given the underlying multistate survival process, is the Landmark Aalen-Johansen estimator [37], which estimates the

expectation of $l_k(t)$ in the landmark cohort of individuals in which calibration is being assessed. The resulting pseudo-values are a vector with $K$ elements, one for each possible transition, for every individual $i$. These pseudo-values can replace the outcome $l_k(t)$ in equations (1) and (2) in order to estimate $o_{j,k}(s,t)$.

Pseudo-values are based on the same assumptions as the underlying estimator $\hat{\theta}$. The Landmark Aalen-Johansen estimator is valid for both Markov and non-Markov multistate models. However, it does make the assumption of random censoring. According to Kleinbaum and Klein [38], this means that "subjects who are censored at time $t$ should be representative of all the study subjects who remained at risk at time $t$ with respect to their survival experience". The approach to alleviate this is to estimate the pseudo-values within sub-groups of individuals, now making the assumption of random censoring within the specified subgroups. This can be done by calculating the pseudo-values within subgroups defined by baseline predictors, or subgroups defined by the predicted transition probabilities $\hat{p}_{j,k}(s,t)$. Both options are implemented in this package. When pseudo-values are calculated within subgroups, they are still used as the outcome in models (1) and (2) in the same way. Note that the pseudo-values $\hat{\theta}^i$ are continuous, as opposed to binary $l_k(t)$, but the link function in model (1) remains the same to ensure $\hat{o}_{j,k}(s,t)$ are between zero and one. Here, $\hat{o}_{j,k}(s,t)$ are commonly referred to as 'pseudo-observed risk' values.

## 2.6 Estimation of confidence intervals

Confidence intervals for both BLR-IPCW and pseudo-value calibration curves can be estimated using bootstrapping. While theoretically feasible, it is currently unclear how to present confidence intervals for each data point in the calibration scatter plots produced by MLR-IPCW, and therefore these are omitted. A process for estimating the confidence intervals around the BLR-IPCW calibration curves is as follows:

1. Resample validation dataset with replacement

2. Landmark the dataset for assessment of calibration

3. Calculate inverse probability of censoring weights

4. Fit the preferred calibration model in the landmarked dataset (restricted cubic splines or loess smoother)

5. Generate observed event probabilities for a fixed vector of predicted transition probabilities (specifically the predicted transition probabilities from the non-bootstrapped landmark validation dataset)

This will produce a number of bootstrapped calibration curves, all plotted over the same vectors of predicted transition probabilities. Taking the $\frac{\alpha}{2}$ and $\left(1 - \frac{\alpha}{2}\right)$ percentiles of the observed event probabilities for each predicted transition probability gives the required $1 - \alpha$ confidence interval around the estimated calibration curve. To estimate confidence intervals for the pseudo-value calibration curves using bootstrapping, the same procedure is applied except the third step is replaced with 'calculate the pseudo-values within the landmarked bootstrapped dataset'. This will be highly computationally demanding as the pseudo-values must be estimated in every bootstrap dataset. This process is integrated into **calibmsm** internally, however code for how to implement these steps manually is also provided in the vignette BLR-IPCW-manual-boostrap [27].

If using the pseudo-value method, confidence intervals can however be calculated using closed form estimates of the standard error when making predictions of the observed event probabilities (i.e., when obtaining fitted values from models (1) and (2)). We recommended this due to the computational burden of bootstrapping the confidence intervals around the pseudo-value calibration curves. There are a number of issues with estimating parametric confidence intervals for the BLR-IPCW calibration curves. Firstly, a robust sandwich-type estimator should be used to estimate the standard error [39], which are known to result in conservative confidence intervals. On the contrary, the size of the confidence interval will be underestimated as uncertainty in estimation of the weights is not considered. Due to the impact of these two factors, we recommend using bootstrapping to estimate the confidence intervals for BLR-IPCW calibration curves.

## 3. Description of package functions and interface

The methods for estimating the calibration curves are implemented using the `calib_msm` function. A step-by-step guide to this process is given in Fig 2.

Step 1 is to obtain access to the multistate model that is being evaluated. **calibmsm** is designed to assess the calibration of a pre-existing model. Details on development of multistate models is given elsewhere [2,15,16].

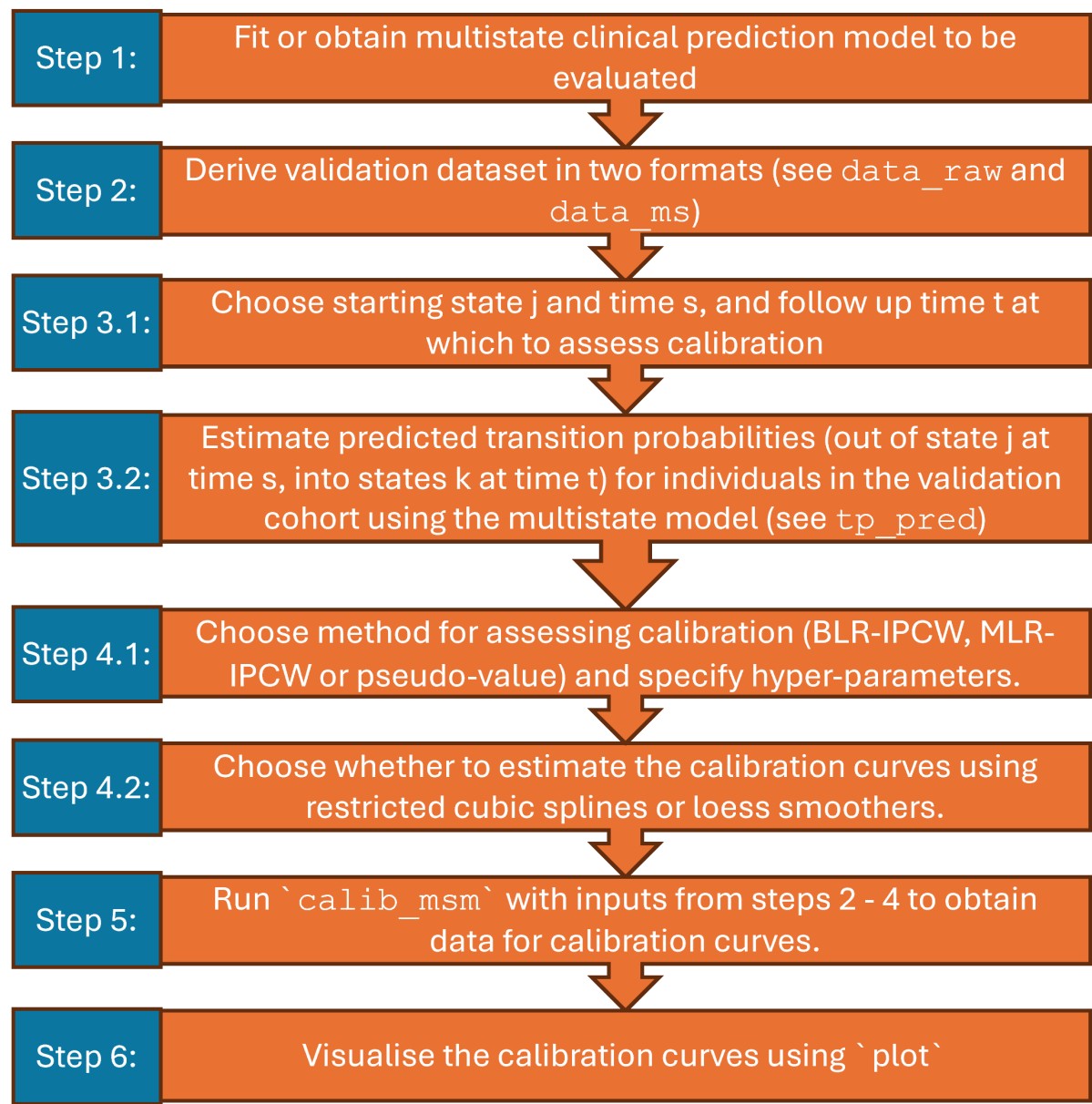

**Fig 2. Flowchart detailing steps for estimating calibration plots using calibmsm::calib_msm.**

Step 2 is to derive the validation dataset that calibration will be assessed in. This must be formatted in two ways, and will be used as the inputs for the `calib_msm` arguments `data_raw` and `data_ms`. The `data_raw` argument requires the validation cohort in a `data.frame` format with one row per individual. For methods BLR-IPCW and MLR-IPCW, `data_raw` must contain variables `dtcens` (censoring time) and `dtcens_s` (censoring indicator), where `dtcens_s = 1` if the individual is censored at time `dtcens`, `dtcens_s = 0` otherwise, plus any baseline predictors **Z** used to estimate the weights. For the pseudo-value approach, `data_raw` must contain any baseline predictors **Z** being used to group individuals before calculating the pseudo-values. The `data_ms` argument requires a dataset of class `msdata`. A dataset of this class can be derived using the package *mstate* [15], and will contain a separate row for every transition for each individual. Both `data_ms` and `data_raw` should contain a corresponding patient ID variable `id`. The `data_raw` and `data_ms` arguments contain the same information about individuals in the validation cohort in different formats, and `data_ms` can be derived from `data_raw` using the function `msprep` from *mstate* [15]. The `data_raw` argument is used to build the calibration models and estimate the calibration curves, while the `data_ms` argument is used to implement the landmarking for all methods, and estimate the Aalen-Johansen estimator for the pseudo-value approach. They are specified separately to reduce computation time when bootstrapping. See section 4.1 for an example of datasets meeting these criteria and details on how they are derived.

Step 3 is to choose the state *j* and time *s* at which calibration will be assessed. The predicted transition probabilities out of state *j* at time *s* for individuals in the validation cohort must then be estimated using the multistate model that is being evaluated. We defer to de Wreede et al [15] and Jackson [16] for details on how to do this. The estimated transition probabilities are then specified in `calib_msm` through the `tp_pred` argument, which must contain a column for each transition *k*, even if the transition from *j* to *k* has zero probability. The rows in `tp_pred` must be ordered in the same way as those in `data_raw`.

Step 4 is to choose a method for assessing calibration. When running `calib_msm`, the user can specify the method for assessing calibration through the `calib_type` argument, and the type of non-linear curve through `curve_type` argument. It is also recommended to specify other relevant input parameters, such as the variables for grouping individuals if using the pseudo-value approach, or the number of knots if using restricted cubic splines. A list of the function arguments and their required inputs is given in Table 1. For full details on function inputs we refer to the package documentation, obtained through `help(package = "calibmsm")`. For users new to R, we recommend starting with the book 'R for Data Science' [41].

The methods in **calibmsm** require continuously observed data, however are agnostic to the type of multistate model used to estimate the transition probabilities. This includes Markov, Semi-Markov or non-Markov models, and non-parametric, semi-parametric or parametric models. A dataset of class `msdata` from *mstate* is required as input, however this is only required to apply landmarking, and determine the occupied state for each individual at time *t*. The estimated transition probabilities, supplied through `tp_pred` can be estimated using any statistical software.

Once the data for the calibration curves or scatter plots has been estimated using `calib_msm`, they can be plotted using the S3 generic, `plot`. Plot methods have been written for three output classes of `calib_msm`: `calib_blr`, `calib_mlr` and `calib_pv`. Separating these processes allows users to manually estimate bootstrapped calibration curves (see vignette BLR-IPCW-manual-bootstrap) [27] using the output from `calib_msm`. It also allows users the flexibility of producing their own plots utilising the full functionality of *ggplot2*, rather than being reliant on the S3 generics provided.

## 4. Clinical example and typical program run

### 4.1 Clinical setting and data preperation

We utilise data from the European Society for Blood and Marrow Transplantation [28], containing multistate survival data after a transplant for patients with blood cancer. The start of follow up is the day of the transplant and the initial state

**Table 1. Arguments for the function calibmsm::calib_msm and required input.**

| Argument/prefix* | Input |
|---|---|
| data_raw | Validation dataset, data.frame, one row per individual. Must contain variables id, dtcens, dtcens_s, and any baseline variables used for estimation of the weights or the grouping of individuals before calculating pseudo-values. |
| data_ms | Validation dataset in msdata format. Must contain variable id, and variables denoting transition history (from, to, trans, Tstart, Tstop, time, status). Created using msdata function from the **mstate** package. Please refer to **mstate** package documentation for more details. |
| j | State from which predicted transition probabilities are made. |
| s | Time from which predicted transition probabilities are made. |
| t | Follow up time at which predicted transition probabilities are made. |
| tp_pred | Predicted transition probabilities, data.frame, one row per individual, one column per state, even if the predicted probabilities into that state are 0. |
| tp_pred_plot | Transition probabilities over which to produce calibration curves, data.frame, one column per state, even if the predicted probabilities into that state are 0. |
| calib_type | Method for estimating the calibration curves or scatter plot. |
| curve_type | Method for modelling the non-linear component of the calibration curves. |
| rcs_nk | Number of knots if estimating the non-linear component using restricted cubic splines. |
| loess_* | Arguments relating to the estimation of the non-linear component using loess smoothers. See stats::loess. |
| mlr_* | Arguments relating to the specification of the vector spline smoother when estimating calibration scatter plots (MLR-IPCW). See VGAM [40]. |
| weights | User-inputted vector of inverse probability of censoring weights. |
| w_* | Arguments relating to the internal estimation of inverse probability of censoring weights. |
| pv_* | Arguments relating to the estimation of the pseudo-values. |
| CI* | Arguments relating to the estimation of a confidence interval. |
| transitions_out | Vector of states that can be transitioned into to produce calibration curves for. Default is to produce calibration curves for any state that can be reached from state j at time s. |
| assess_moderate | Whether to assess moderate calibration |
| assess_mean | Whether to assess mean calibration. |

*Arguments that correspond to the same aspect of the process have been grouped. For details on each individual argument please refer to the package documentation [27].

is alive and in remission. There are three intermediate events ($2$: recovery, $3$: adverse event, or $4$: recovery + adverse event), and two absorbing states ($5$: relapse and $6$: death). This data is available from the **mstate** package. We assume the user of **calibmsm** has experience with handling the type of data used to develop a multistate model as outlined by De Wreede et al [15].

Four datasets are provided to enable assessment of a multistate model fitted to these data. The code for deriving all these datasets is provided in the source code for **calibmsm** [27]. The first is ebmtcal, which is the input for argument data_raw. This data is the same as the ebmt dataset provided in **mstate**, with two extra variables derived: time until censoring (dtcens) and an indicator for whether censoring was observed (dtcens_s = 1) or an absorbing state was entered (dtcens_s = 0). This dataset contains baseline information on year of transplant (year), age at transplant (age), prophylaxis given (proph), and whether the donor was gender matched (match), and information on time until entry into the different outcome states. Such a dataset would need to be constructed by the user after prospective data collection or manipulation of observational data.

The second dataset provided is msebmtcal, which is the input for argument data_ms. This is a dataset of class msdata, and has been derived from the ebmt dataset by applying the processes and functions from the package

*mstate* [15]. It contains all transition times, an event indicator for each transition, as well as a `trans` attribute containing the transition matrix. Note that while the data on transition times is not required to be present in `ebmtcal` in order to run `calib_msm`, it is required in order to derive the dataset `msebmtcal` from `ebmtcal`.

```
library(calibmsm)

data("ebmtcal")
head(ebmtcal, n = 3)

##   id  rec rec.s  ae ae.s recae recae.s  rel rel.s  srv srv.s      year
agecl
## 1  1   22     1 995    0   995       0  995     0  995     0 1995-1998
20-40
## 2  2   29     1  12    1    29       1  422     1  579     1 1995-1998
20-40
## 3  3 1264     0  27    1  1264       0 1264     0 1264     0 1995-1998
20-40
##   proph                  match dtcens dtcens_s
## 1    no no gender mismatch    995        1
## 2    no no gender mismatch    422        0
## 3    no no gender mismatch   1264        1

data("msebmtcal")
head(msebmtcal, n = 10)

##    id from to trans Tstart Tstop time status
## 1   1    1  2     1      0    22   22      1
## 2   1    1  3     2      0    22   22      0
## 3   1    1  5     3      0    22   22      0
## 4   1    1  6     4      0    22   22      0
## 5   1    2  4     5     22   995  973      0
## 6   1    2  5     6     22   995  973      0
## 7   1    2  6     7     22   995  973      0
## 8   2    1  2     1      0    12   12      0
## 9   2    1  3     2      0    12   12      1
## 10  2    1  5     3      0    12   12      0
```

In the work of De Wreede et al. [15], the focus is on predicting transition probabilities made at times $s = 0$ and $s = 100$ days, across a range of follow up times $t$, and comparing prognosis for patients in different states $j$. In this study we also focus on assessing the calibration of the transition probabilities made at these times. We assess calibration of the transition probabilities at $t = 5$ years, a common follow up time for cancer prognosis, but calibration of the model may vary for other values of $t$. We estimate transition probabilities for each individual by developing a model as demonstrated in de Wreede et al. [15], following the theory of Putter et al [2].

The predicted transitions probabilities from each state $j$ at times $s = 0$ and $s = 100$ are contained in stacked datasets `tps0` and `tps100` respectively. A leave-one-out approach was used when estimating these transition probabilities. This means each individual was removed from the development dataset when fitting the multistate model to estimate their transition probabilities. This approach allows validation to be assessed in the same dataset that the model was developed with minimal levels of in-sample optimism. Note that for `tps100` the predicted probabilities for some states $k$ are equal to $0$. This is because no individuals in state $j = 1$ at time $s = 100$ transition into states $3$ or $4$. This may be due to the definition of an adverse event having to occur within a certain number of days post transplant.

```
data("tps0")
head(tps0, n = 3)

##   id   pstate1   pstate2    pstate3   pstate4   pstate5   pstate6 se1
## 1  1 0.1139726 0.2295006 0.08450376 0.2326861 0.1504855 0.1888514 0.012
91133
## 2  2 0.1140189 0.2316569 0.08442692 0.2328398 0.1481977 0.1888598 0.012
91552
## 3  3 0.1136646 0.2317636 0.08274331 0.2325663 0.1504787 0.1887834 0.012
89444
##          se2        se3        se4        se5        se6 j
## 1 0.02369584 0.01257251 0.02323376 0.01648630 0.01601795 1
## 2 0.02374329 0.01256056 0.02324869 0.01632797 0.01603703 1
## 3 0.02375770 0.01245752 0.02322375 0.01647890 0.01601525 1

data("tps100")
head(tps100, n = 3)

##   id   pstate1    pstate2 pstate3 pstate4   pstate5   pstate6       se
1
## 1  1 0.7013881 0.05239271       0       0 0.1408120 0.1054072 0.0469116
8
## 2  2 0.7012745 0.05261136       0       0 0.1407625 0.1053516 0.0469121
8
## 3  3 0.7011368 0.05270176       0       0 0.1407628 0.1053987 0.0469306
8
##          se2 se3 se4        se5        se6 j
## 1 0.02077138   0   0 0.03457006 0.03081258 1
## 2 0.02082871   0   0 0.03456448 0.03079617 1
## 3 0.02086917   0   0 0.03456101 0.03081033 1
```

### 4.2 Calibration plots for the transition probabilities out of state $j = 1$ at time $s = 0$

We start by producing calibration curves for the predicted transition probabilities out of state $j = 1$ at time $s = 0$. Given all individuals start in state $1$, there is no need to consider the transition probabilities out of states $j \neq 1$ at $s = 0$. Calibration is assessed at follow up time ($t = 1826$ days). We start by extracting the predicted transition probabilities from state $j = 1$ at time $s = 0$ from the object `tps0`. These are the transition probabilities we aim to assess the calibration of.

```
tp_pred_s0 <- tps0 |>
  dplyr::filter(j == 1) |>
  dplyr::select(any_of(paste("pstate", 1:6, sep = "")))
```

We first evaluate calibration using the BLR-IPCW approach by specifying `calib_type = "blr"`. We choose to estimate the calibration curves using restricted cubic splines, although the use of loess smoothers would be equally valid. When using restricted cubic splines, the number of knots must always be specified by the user, and 3 knots are chosen here given the reasonably small size of the dataset. Weights are estimated using the internal estimation procedure with baseline predictor variables `year`, `agecl`, `proph` and `match`. The `w_max_follow=t_eval` argument censors individuals at `t_eval` before fitting the model used to estimate the weights, i.e., a "stopped cox" approach [42]. This decision was made to help meet the proportional hazards assumption as there is differential follow up for individuals in different `year` groups (see vignette Sensitivity-analysis-for-IPCWs [27] for more details). The `w_land-mark_type` argument assigns whether weights are estimated using all individuals uncensored at time $s$, or only those uncensored and in state $j$ at time $s$, as discussed in section 2.4 The maximum weight (`w_max = 10`) and stabilisation of weights (`w_stabilised = TRUE`) are left as default. Weights can also be manually specified using the `weights` argument. We request 95% confidence intervals for the calibration curves calculated through bootstrapping with 200 bootstrap replicates.

```
t_eval <- 1826

dat_calib_blr <-
  calib_msm(data_ms = msebmtcal,
            data_raw = ebmtcal,
            j=1,
            s=0,
            t = t_eval,
            tp_pred = tp_pred_s0,
            calib_type = "blr",
            curve_type = "rcs",
            rcs_nk = 3,
            w_covs = c("year", "agecl", "proph", "match"),
            CI = 95,
            CI_R_boot = 200)
```

The first element of `dat_calib_blr` (named `plotdata`) contains 6 data frames. One for the calibration curves of the transition probabilities into each of the six states, $k \in \{1, 2, 3, 4, 5, 6\}$. Each data frame contains five columns, `id`: the identifier of each individual; `pred`: the predicted transition probabilities; `obs`: the observed event probabilities; `obs_lower` and `obs_upper`: the confidence interval for the observed event probabilities. The second element (named `metadata`) is a metadata argument containing information about the data and chosen calibration analysis. The plot data and metadata can be viewed using the `print` and `metadata` commands respectively. However, it is recommended to get acquainted with the underlying object structure, as accessing the plot data will be useful if wanting to customise plots or apply bootstrapping manually.

```
print(dat_calib_blr)

## $state1
##   id       pred         obs  obs_lower obs_upper
## 2  2 0.1140189 0.1095897 0.08797538 0.1331918
## 4  4 0.1383878 0.1036308 0.08151624 0.1239493
## 5  5 0.1233226 0.1051035 0.08563362 0.1247074
##
## $state2
##   id       pred         obs obs_lower obs_upper
## 2  2 0.2316569 0.1698031 0.1195364 0.2224530
## 4  4 0.1836189 0.1855591 0.1588683 0.2141211
## 5  5 0.1609740 0.1759804 0.1456545 0.2051705
##
## $state3
##   id       pred         obs  obs_lower obs_upper
## 2  2 0.08442692 0.12485834 0.08999565 0.1544482
## 4  4 0.07579429 0.11666056 0.08333101 0.1446896
## 5  5 0.05508100 0.09189341 0.04947315 0.1378489
##
## $state4
##   id       pred         obs obs_lower obs_upper
## 2  2 0.2328398 0.2427580 0.1998414 0.2853412
## 4  4 0.2179331 0.2243106 0.1911869 0.2580538
## 5  5 0.1828176 0.1851051 0.1543974 0.2150953
##
## $state5
##   id       pred         obs obs_lower obs_upper
## 2  2 0.1481977 0.1909795 0.1668994 0.2206078
## 4  4 0.1538475 0.1654523 0.1502043 0.1829148
## 5  5 0.1425950 0.2215190 0.1785118 0.2722747
##
## $state6
##   id       pred         obs obs_lower obs_upper
## 2  2 0.1888598 0.2069354 0.1850181 0.2302357
## 4  4 0.2304185 0.2542212 0.2235304 0.2829786
## 5  5 0.3352099 0.3163102 0.2792628 0.3520205

metadata(dat_calib_blr)

## $valid_transitions
## [1] 1 2 3 4 5 6
##
## $assessed_transitions
## [1] 1 2 3 4 5 6
##
## $CI
## [1] 95
##
## $CI_type
## [1] "bootstrap"
##
## $CI_R_boot
## [1] 200
##
## $j
## [1] 1
##
## $s
## [1] 0
##
## $t
## [1] 1826
##
## $calib_type
## [1] "blr"
##
## $curve_type
## [1] "rcs"
```

The BLR-IPCW calibration curves can then be generated by applying the `plot` function to the output from *calib_msm*. Note, when marginal density curves are requested (default), *grid::grid_draw* is the preferred approach to display this plot in the viewer. The calibration curves (Fig 3) indicate the level of calibration is different for the transition probabilities into each of the different states. The calibration into states 4 and 6 looks the best. State 2 has good calibration over the majority of the predicted risks but over predicts for individuals with the highest predicted risks. Transition probabilities into states 1 and 3 are over and under predicted respectively over most of the range of predicted risks. Importantly the calibration of the transition probabilities into state 5 (Relapse), a key clinical outcome in this clinical setting, is extremely poor.

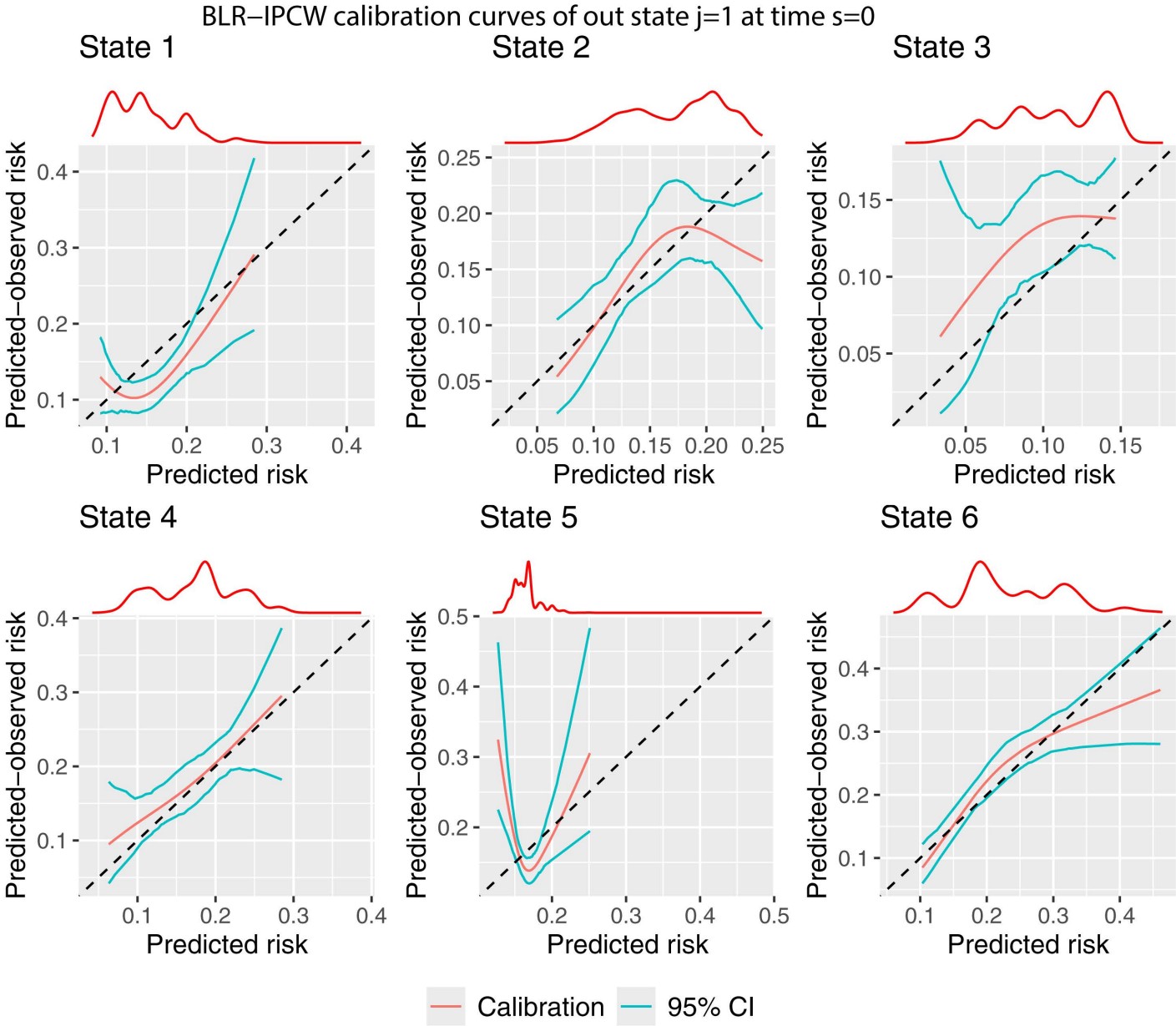

**Fig 3. BLR-IPCW calibration curves out of state j = 1 at time s = 0.**

```
plot_pv <- plot(dat_calib_pv, combine = TRUE, nrow = 2, ncol = 3)
grid::grid.draw(plot_pv)
```

Next we use the pseudo-value approach to assess calibration by specifying `calib_type = "pv"`. Instead of specifying how the weights are estimated, we now specify variables to define groups within which pseudo-values will be calculated (see section 2.5). The goal is to induce uninformative censoring within the chosen subgroups. We chose to calculate pseudo-values in individuals with the same year of transplant (`pv_group_vars = c("year")`), and then split individuals into a further three groups defined by their predicted risk (`pv_n_pctls = 3`). The number of percentiles should be increased in bigger validation datasets, although guidance on specific numbers is currently lacking. Year of transplant was identified as a subgrouping variable because a later transplant resulted in a shorter possible follow up, an earlier administrative censoring time, and it was therefore highly predictive of being censored. Your data should be explored to identify appropriate variables for subgrouping (see vignette Evaluation-of-estimation-of-IPCWs). A parametric confidence interval is estimated as recommended in section 2.6.

```
dat_calib_pv <-
  calib_msm(data_ms = msebmtcal,
          data_raw = ebmtcal,
          j = 1,
          s = 0,
          t = t_eval,
          tp_pred = tp_pred_s0,
          calib_type = "pv",
          curve_type = "rcs",
          rcs_nk = 3,
          pv_group_vars = c("year"),
          pv_n_pctls = 3,
          CI = 95,
          CI_type = "parametric")

plot_pv <- plot(dat_calib_pv, combine = TRUE, nrow = 2, ncol = 3)
grid::grid.draw(plot_pv)
```

The pseudo-value calibration curves (Fig 4) are largely similar to the BLR-IPCW calibration curves (Fig 3). The agreement in the calibration curves from two completely distinct methods provides some reassurance on the validity of the assessment of calibration. This is with the exception of state $k = 3$, where the pseudo-value calibration plot indicates the transition probabilities are well calibrated, but the BLR-IPCW calibration plot indicates the transition probabilities under predict. In a situation like this, we recommend testing the assumptions made by each of the methods to try and diagnose which are most likely to hold, and what may be driving the difference. In this particular example, we hypothesised that the cox-proportional hazards model for estimating the inverse probability of censoring weights may be misspecified, in particular the proportional hazards assumption, due to the strong effect of year of transplant on the censoring mechanism. We explored this theory in more detail (see vignette Sensitivity-analysis-for-IPCWs), but found little change when estimating the weights using a flexible parametric survival model. Instead, we identified that this may be caused by a difference in the censoring mechanism for individuals in the adverse event state, as it appeared these individuals were less likely to be censored. This will bias the results from the BLR-IPCW and MLR-IPCW methods unless the weights are conditional on the amount of time spent in each outcome state, something which **calibmsm** is not currently set up to do. Although it's not possible to be certain that the individuals in the adverse event state were less likely to be censored purely from looking at the data, we concluded it was a strong possibility, and that the BLR-IPCW calibration curves may be biased in this particular clinical example.

Next we use the MLR-IPCW to evaluate calibration which produces a calibration scatter plot by specifying `calib_type = "mlr"`. The inputs for calculating the weights are the same as for the BLR-IPCW approach, but a confidence interval is no longer requested which is not possible for the MLR-IPCW approach.

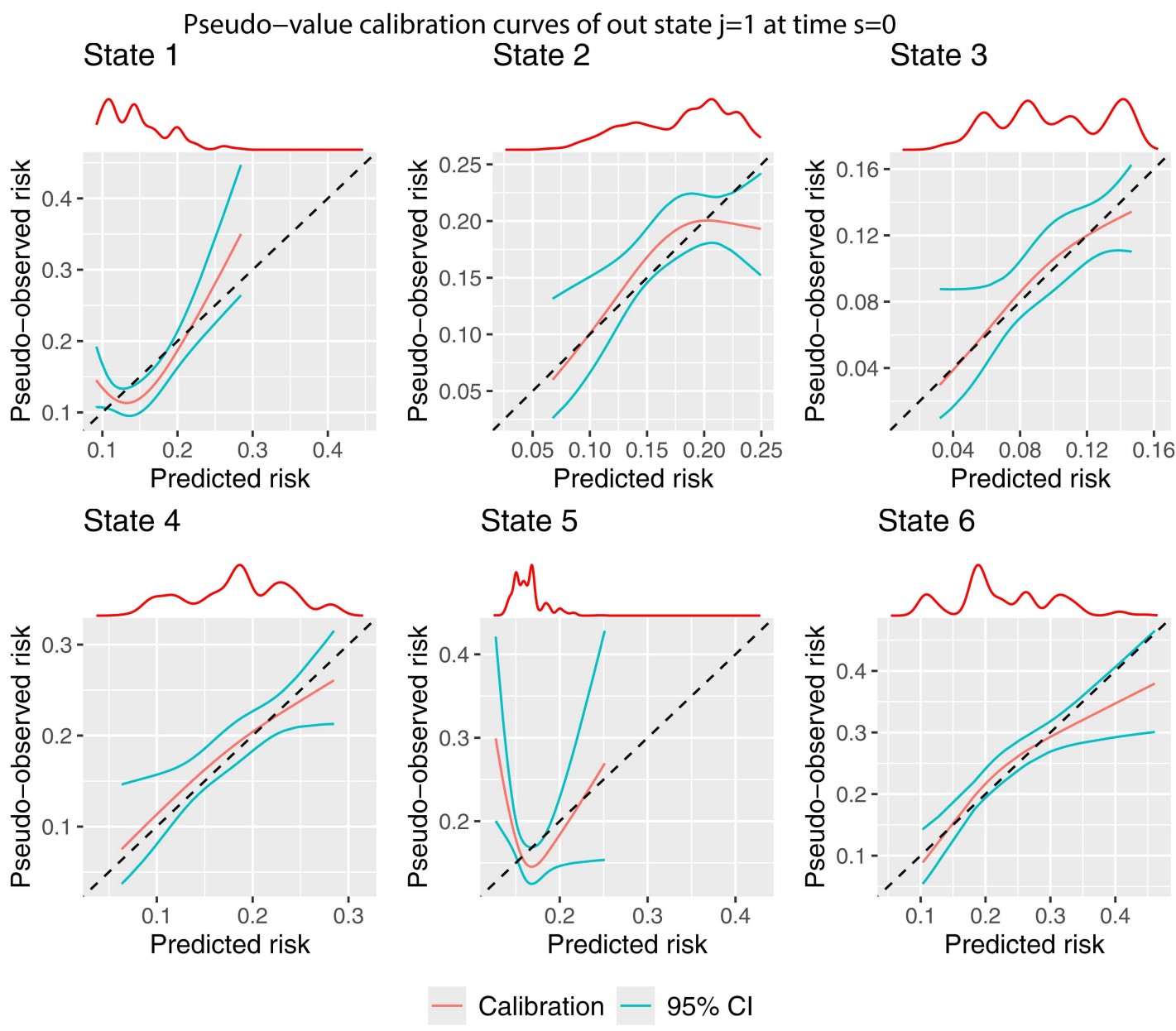

**Fig 4. Pseudo-value calibration curves out of state j = 1 at time s = 0.**

```
dat_calib_mlr <-
  calib_msm(data_ms = msebmtcal,
            data_raw = ebmtcal,
            j = 1,
            s= 0,
            t = 1826,
            tp_pred = tp_pred_s0,
            calib_type = "mlr",
            w_covs = c("year", "agecl", "proph", "match"))

plot_mlr <- plot(dat_calib_mlr, combine = TRUE, nrow = 2, ncol = 3)
grid::grid.draw(plot_mlr)
```

The MLR-IPCW calibration scatter plots, produced using `plot` are contained in Fig 5. Within each plot for state $k$, there is a large amount of variation in calibration of the transition probabilities depending on the predicted transition probabilities into states $\neq k$. One valuable insight from these plots is that the variance in the calibration of the transition probabilities into state $6$, is considerably smaller than that of state $4$, despite these two states both having good calibration according to the BLR-IPCW plots (arguably state $4$ looked better calibrated). This means the calibration of the transition probabilities into state $6$ remains reasonably consistent, irrespective of the risks of the other states. On the contrary, the calibration of the predicted transition probabilities into state $4$ is more dependent on the predicted transition probabilities of the other states. These plots can also be used to identify groups of individuals where calibration is poor. For example, there is a cluster of individuals with a predicted-observed risk > 17.5% for state 3, where the model is under-predicting. These are all individuals aged 20–40, year group 1995–1998, no gender mismatch and no prophylaxis.

In practice, we recommend researchers produce calibration curves using all three methods. The MLR-IPCW approach is a stronger [9] form of calibration assessment than the BLR-IPCW and pseudo-value approaches, but all

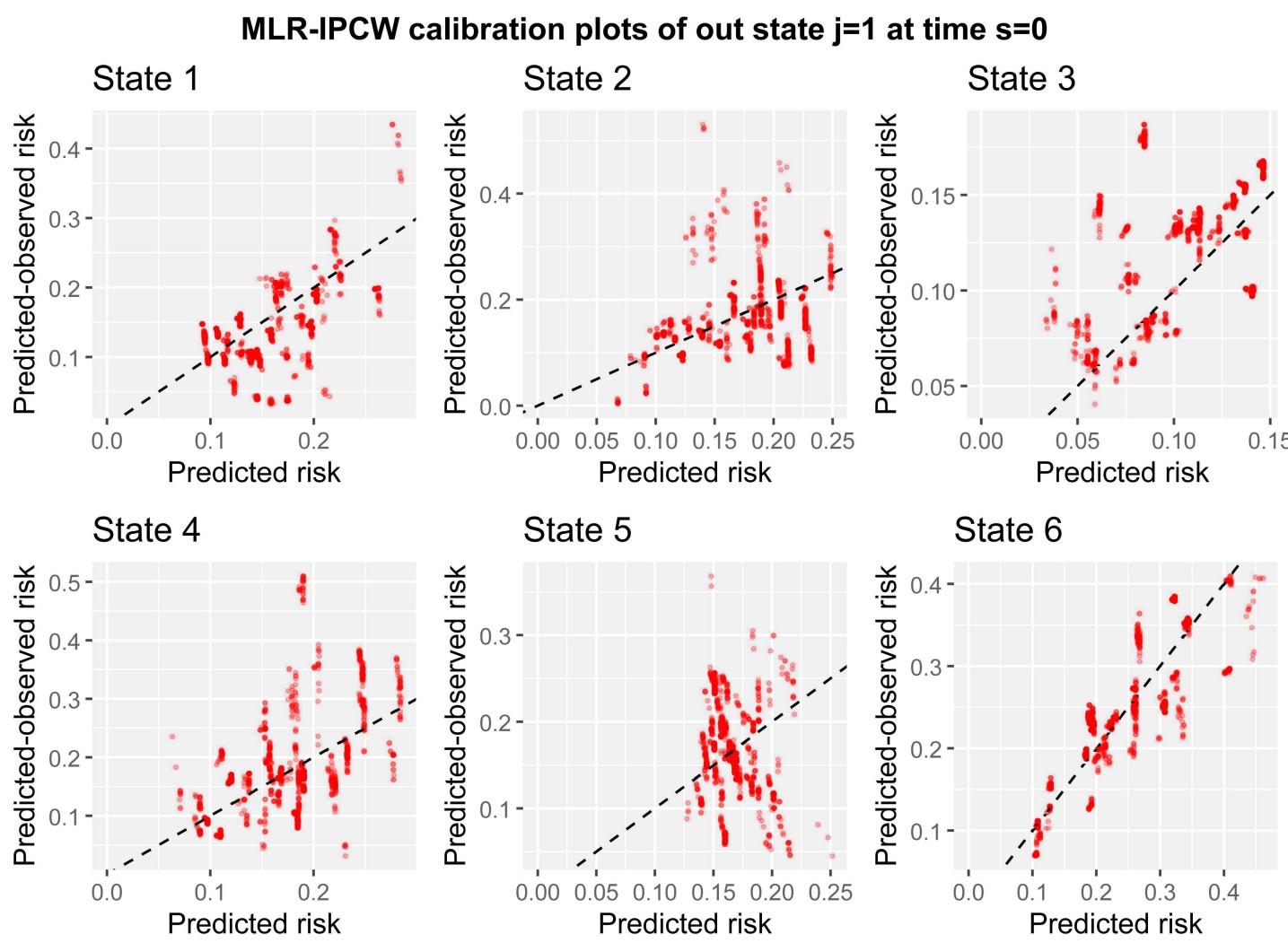

**Fig 5. MLR-IPCW calibration scatter plots out of state j = 1 at time s = 0.**

three provide important contextual information. Note that we hypothesized that the inverse probability of censoring weights may not be accurate in this example due to a censoring mechanism which changes depending on outcome state occupancy. Using the current approach for estimating the weights, this will result in biased calibration plots for the BLR-IPCW and MLR-IPCW methods, having a particular effect on the plots for state 3 (adverse event). For the following section, we therefore proceed using only the pseudo-value method.

### 4.3 Calibration plots for the transition probabilities out of states $j = 1$ and $3$ at time $s = 100$

In the work of De Wreede et al. [15], focus then shifts to comparing transition probabilities when $s = 100$ depending on whether an individual has had an adverse event (state $3$) or remains in state $1$ (post transplant). Our focus therefore now shifts to assessing the calibration of these transition probabilities. This is done through landmarking as described in section 2. We start by extracting the predicted transition probabilities from state $j = 1$ and $3$ at time $s = 100$ from the object `tps100`. These are the transition probabilities we aim to assess the calibration of.

```
tp_pred_j1s100 <- tps100 |>
                dplyr::filter(j == 1) |>
                dplyr::select(any_of(paste("pstate", 1:6, sep = "")))

tp_pred_j3s100 <- tps100 |>
                dplyr::filter(j == 3) |>
                dplyr::select(any_of(paste("pstate", 1:6, sep = "")))
```

The process for estimating the calibration curves remains the same, changing the inputted values `j` and `s`, and specifying the appropriate predicted transition probabilities to the argument `tp_pred`. We start by producing the calibration plots for $j = 1$ and $s = 100$ using pseudo-value (Fig 6) method.

```
dat_calib_pv_j1_s100 <-
  calib_msm(data_ms = msebmtcal,
          data_raw = ebmtcal,
          j = 1,
          s = 100,
          t = t_eval,
          tp_pred = tp_pred_j1s100,
          calib_type = "pv",
          curve_type = "rcs",
          rcs_nk = 3,
          pv_group_vars = c("year"),
          CI = 95,
          CI_type = "parametric")
plot_blr_j1_s100 <- plot(dat_calib_blr_j1_s100, combine = TRUE, nrow = 2,
ncol = 3)
grid::grid.draw(plot_blr_j1_s100)
```

There are only four calibration plots because no individuals in state $j = 1$ at time $s = 100$ are in states $k = 3$ (adverse event) or $k = 4$ (recovery + adverse event) after $t = 1826$ days. We believe this is due to the definition of an adverse event occuring within 100 days, but as secondary users of the data, cannot be sure about this. The calibration of the predicted transition probabilities according is very poor. Only for state $k = 6$ is the observed risk a monotonically

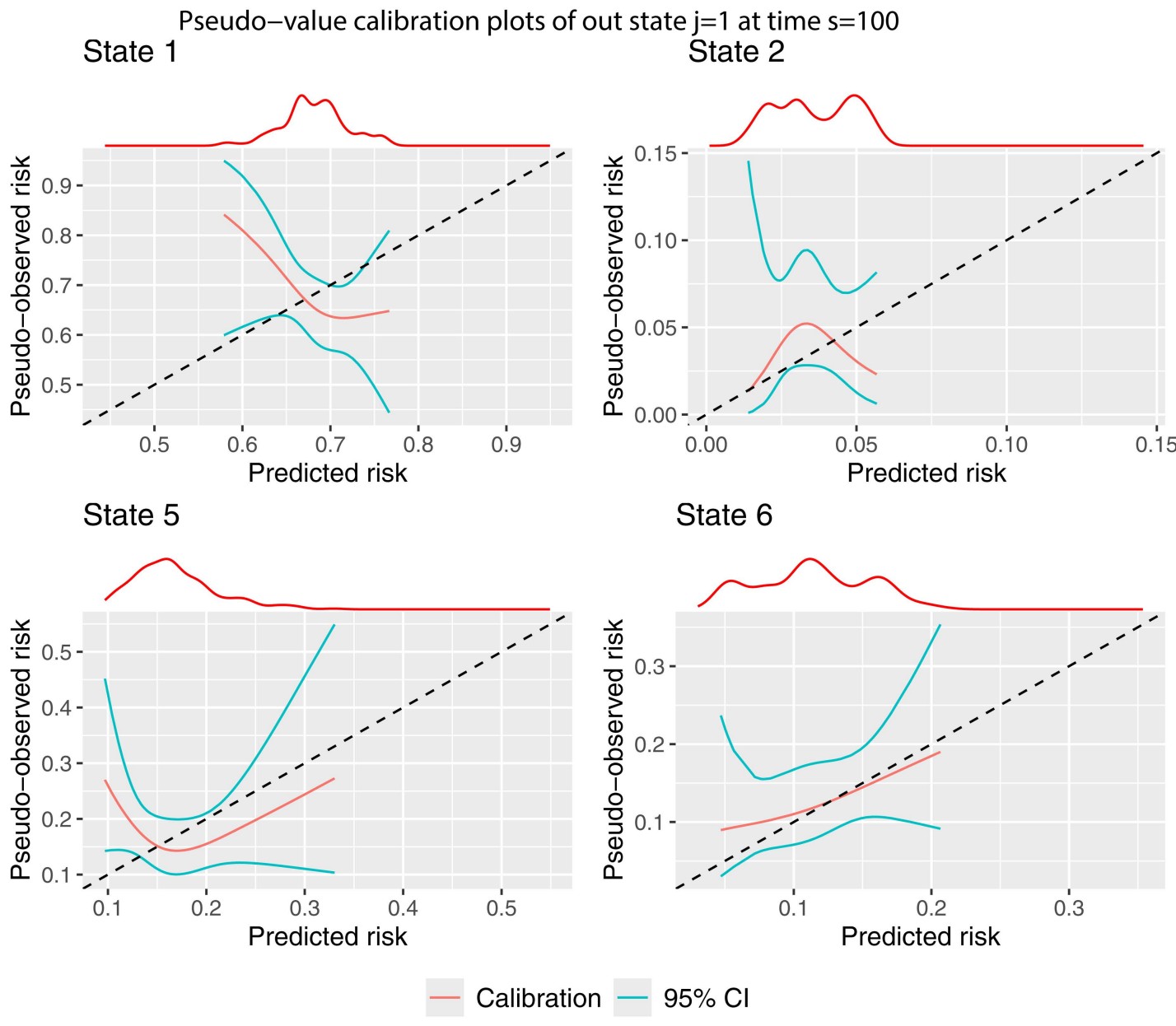

**Fig 6. Pseudo-value calibration curves out of state j = 1 at time s = 100.**

increasing function of the predicted transition probabilities. The confidence intervals are very large. For states $k = 2$ and $k = 5$, we cannot rule out that the poor calibration is a result of sampling variation as opposed to a poorly performing prediction model. A larger validation dataset would be required to get to the bottom of this. There is a major issue with the calibration of the transition probabilities of staying in state $1$, as the predicted risk is inversely proportional to the observed event rate.

Next we produce calibration plots for $j = 3$ and $s = 100$ (Fig 7).

```
dat_calib_pv_j3_s100 <-
    calib_msm(data_ms = msebmtcal,
              data_raw = ebmtcal,
              j = 1,
              s = 100,
              t = t_eval,
              tp_pred = tp_pred_j1s100,
              calib_type = "pv",
              curve_type = "rcs",
              rcs_nk = 3,
              pv_group_vars = c("year"),
              CI = 95,
              CI_type = "parametric")

plot_blr_j3_s100 <- plot(dat_calib_blr_j3_s100, combine = TRUE, nrow = 2,
ncol = 3)
grid::grid.draw(plot_blr_j3_s100)
```

Again there are only four possible states that an individual may transition into, although this includes states $3$ (adverse event) and $4$ (recovery + adverse event), instead of $1$ (post transplant) and $2$ (recovery). This is because once an individual has entered state $3$, they cannot move backwards into states $1$ or $2$. The calibration plots are better than for $j = 1$. For transitions into states $k = 3,\ 4$ and $6$, the calibration curves are monotonically increasing and comparatively close to the line of perfect calibration, although the confidence intervals are still quite large. Again the calibration of state $5$ is very poor. This makes it difficult to base any clinical decisions on the predicted transition probabilities for relapse out of states $j = 1$ or $3$ at time $s = 100$, whereas making clinical decisions based on the risk of death ($k = 6$) after survival for $100$ days is more viable, as this was well calibrated for both $j = 1$ and $j = 3$.

We provide an overview of interpretating the calibration curves for this clinical example in Box 1.

Box 1. Interpretation of the calibration results.

Please note, the model being evaluated was originally developed to showcase the functionality of the **mstate** package and was never intended for clinical use. This is a hypothetical discussion to illustrate how the calibration plots may be interpreted.

 Assessing the calibration of multi-state clinical prediction models requires consideration of each of the states of the model, with a requirement for the re to be good calibration across all states before the model could be used in clinical practice. We have provided three methods to assess calibration (see Section 2), and we recommend assessing calibration using each, so that the results can be compared.

The calibration curves shown in Figs 3–5, which consider predictions out of the recovery state a time 0, show that there is good agreement between the observed and predicted risks for some – but not all – of the states. The results tell us that the transition probabilities of remaining in state $1$ (transplant) are pre-dominantly over-predicting. Specifically, the model is over-estimating the predicted risk of someone not recovering, having an adverse event, experiencing relapse or death following transplant. The transition probabilities of being in state 2 (recovery) or state 5 (adverse event + recovery) are either under or over-predicting depending on the predicted risk value. A key clinical outcome in this clinical setting is the risk of relapse (state 5), with these results showing that the model should not be used to inform risk estimation for this state, since the calibration of state 5 is extremely poor.

On the contrary, the calibration of transition probabilities for state 4 (adverse event + recovery) and state 6 (death) are reasonably well calibrated. Checking for consistency in conclusions across the three calibration methods is always recommended as it may reveal important insights from the analysis. Indeed, we found differences in the calibration results of state 3 across the three methods (as discussed in the main paper). This led to further investigation, which concluded that the calibration plots from the BLR-IPCW and MLR-IPCW approaches may be biased in this setting, in particular state 3 (adverse event). This led us to focus on the pseudo-value calibration plots which indicated the transition probabilities into the adverse event state were well calibrated.

The pseudo-value calibration curves in Figs 6 and 7, which consider predictions out of the recovery and adverse event states at time 100, show very poor agreement between the observed and predicted risks.

In our opinion, finding that there are some states with miscalibrated transition probabilities informs us that the predicted risks from the model should not be used to inform clinical decision-making. For example, it is clear that the model should not be used to aid clinical decision-making around relapse risk following transplant, especially when making predictions 100 days post-transplant. On the contrary, one may argue that if using the model to inform a clinical decision based solely around the risk of death (state 6), the model is appropriate. However, if the goal is not to make a clinical decision informed by the risk of all the states simultaneously, a multistate model is not necessary in the first place.

From this, one would conclude revisions to the model are needed prior to model implementation. For example, one could test the inclusion of interaction terms between predictors, or one could reduce the number of predictors in specific transition models to reduce overfitting. However, diagnosing which part of the multi-state model is causing certain transition probabilities to be miscalibrated is multifaceted. Indeed, miscalibrated transition probabilities could be driven by errors in any of the intermediate competing risks models into that state. Further research is needed around methods to help such identification.

## 5. Discussion

Multistate models are a unique tool for prediction, handling both competing risks and the occurrence of intermediate health states in the same model. Development of multistate models for prediction is becoming more common, yet validation of such models is still very uncommon. A major barrier to the implementation of statistical techniques is often the availability of software [43]. **calibmsm** has been developed to aid in the implementation of techniques to assess the calibration of the transition probabilities from a multistate model. This paper has extended previously proposed methods for assessing the calibration of the transition probabilities out of the initial state [24], to the transition probabilities out of any state $j$ at any time $s$. While package development has focused on multistate models, **calibmsm** could, in theory, be used to assess the calibration of predicted risks from a range of other models, including: competing risks models [2], standard single outcome survival models, where predictions can be made at any landmark time (note these are both special cases of a multistate model), dynamic models [26,44], and any model which utilises information post baseline to update predictions [45].

All three methods (BLR-IPCW, MLR-IPCW and pseudo-value) have been shown to give an unbiased assessment of calibration under random censoring mechanisms, and a predominately unbiased assessment of calibration when there is a strong association between the outcome and censoring mechanisms that can be explained by baseline covariates [24]. This paper found broadly similar evaluation of calibration when using the BLR-IPCW and pseudo-value methods, however there were discrepancies in the evaluation of calibration of the transition probabilities into state $k = 3$. This is an indicator that the assumptions underpinning either one of the methods could be violated. This was explored in detail

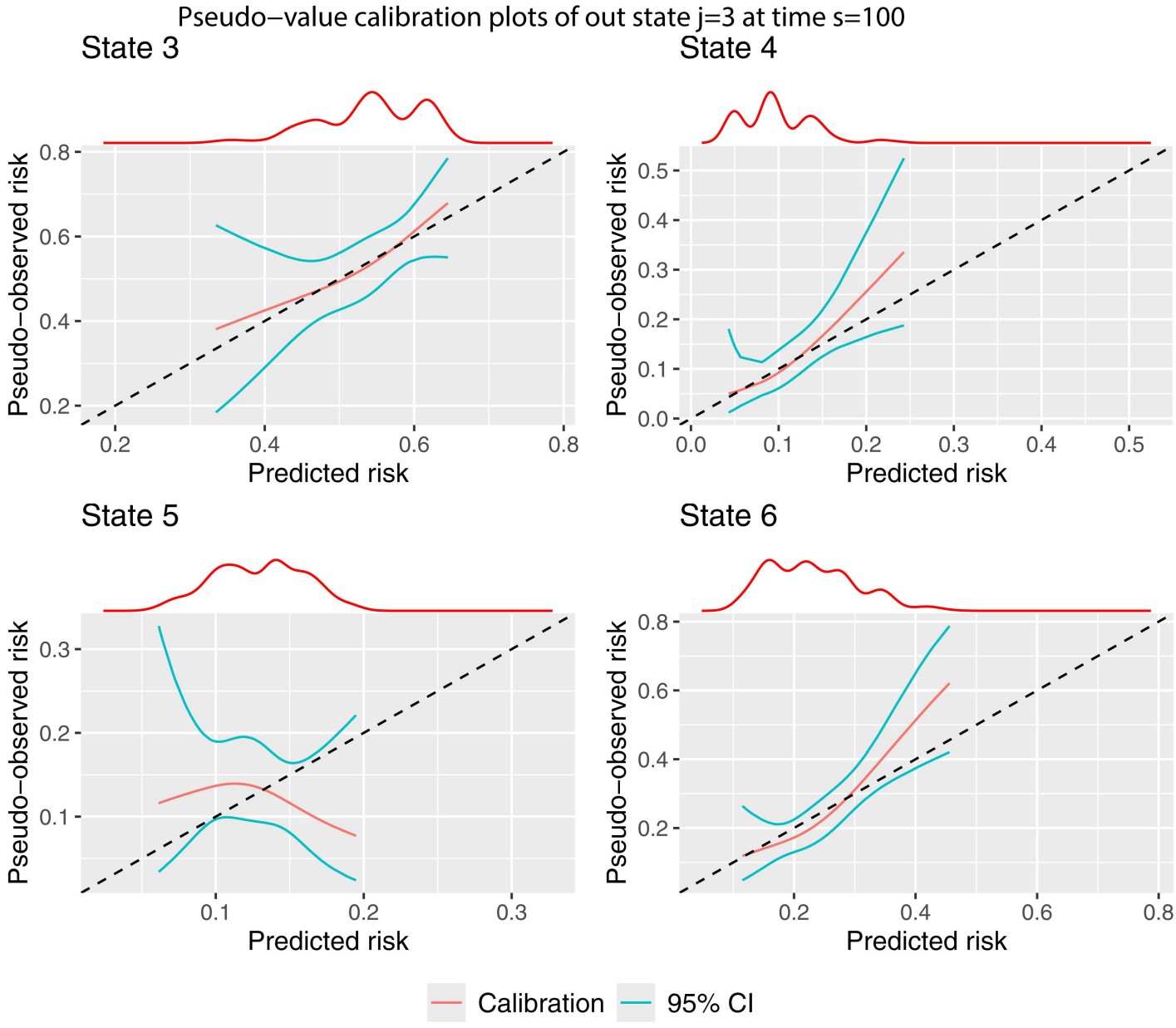

**Fig 7. Pseudo-value calibration curves out of state j = 3 at time s = 100.**

(see vignette Sensitivity-analysis-for-IPCWs [27]) and led to the conclusion that the BLR-IPCW and MLR-IPCW plots are likely unreliable, in particular for the adverse event state. We hypothesised this was driven by a differential censoring mechanism/observation process for individuals in the adverse event state. Simulations studies are required to 1) quantify this type of bias, and 2) explore whether this can be accounted for by estimating the inverse probability of censoring weights using approaches which are conditional on the time spent in each outcome state (for example a latent-class model). If such a study could be undertaken this would be highly valuable [46,47]. For now, we reiterate the importance

of implementing these methods in settings where the observation process/censoring mechanism does not change depending on the outcome state an individual is in. It has previously been suggested to evaluate calibration using MLR-IPCW and one of the BLR-IPCW or pseudo-value approaches because MLR-IPCW provides a stronger assessment of calibration [24]. However, we now suggest to evaluate calibration using all three methods, and a comparison between the BLR-IPCW and pseudo-value approaches can be used to help assess whether the assumptions of either method may be violated.

Given it is possible to use **calibmsm** to validate a standard competing risks model [12,13], we carried out a sensitivity analysis to compare the approaches described in this paper with the 'graphical calibration curves' of Austin et al. [12], (see vignette Comparison-with-graphical-calibration-curves-in-competing-risks-setting [27]). BLR-IPCW, pseudo-values, and graphical calibration curves (MLR-IPCW excluded for not producing a calibration curve) all resulted in similar calibration curves. This is with the exception of BLR-IPCW for state $k = 3$, which has been previously discussed. The three methods take completely different approaches to assessing the calibration of a competing risks model. Therefore finding agreement between these assessments of calibration can provide reassurance that the calibration plots are correct, and is an exercise that could be repeated in practice. Despite this, the relative performance of each method in a wider range of competing risks scenarios remains unknown. A comparison of these methods in a simulation when the assumptions of each method do and do not hold, and under a range of sample sizes and multistate model structures, would be therefore valuable [46].

The BLR-IPCW, MLR-IPCW and pseudo-value approaches have different computational burdens. A calibration curve can be obtained reasonably quickly using the BLR-IPCW or MLR-IPCW approaches, however estimation of confidence intervals for BLR-IPCW using bootstrapping (the recommend method in section 2.6) will result in a high computation time in large validation datasets. On the contrary, obtaining the calibration curve itself using the pseudo-value approach has a high computational burden due to estimation of the pseudo-values. Once these have been calculated, a calibration curve and confidence interval can be estimated quickly using parametric techniques, meaning estimation of the confidence interval adds minimal computational burden. We plan to extend the package to allow users to estimate the pseudo-values for each individual seperately before estimating the calibration curve. This will allow the first part of the process to be parallelised and will make estimation of calibration curves using the pseudo-value approach more feasible in large datasets.

Estimation of the weights is clearly of high importance for the BLR-IPCW and MLR-IPCW approaches. If the model to do so is misspecified, this could lead to incorrect evaluation of the calibration. It is possible this is what is causing the difference between the BLR-IPCW and pseudo-value approaches for the calibration of transition probabilities from state $j = 1$ at time $s = 0$ into state $k = 3$, as was explored in vignette Sensitivity-analysis-for-IPCWs [27]. This package is focused on creation of calibration curves, but is not a dedicated package for estimating inverse probability of censoring weights. We encourage users to create a well specified model for the weights (see Robins and Hernan [39]) if using the BLR-IPCW or MLR-IPCW approaches. Custom functions for estimating the weights can be specified through the `w.function` argument in *calib_msm*. Alternatively, weights can be estimated externally and then specified though the *weights* argument. In this latter case, the internal bootstrapping procedure will not work, as the weights need to be re-estimated in each bootstrap dataset. We have provided a more detailed vignette about how to estimate calibration curves and confidence intervals using bootstrapping when defining your own function to estimate the weights (see vignette BLR-IPCW-manual-bootstrap) [27].

Despite sample size formulae being available for clinical prediction models predicting continuous [48], binary [34,49], time-to-event [34] and multinomial outcomes [50]; sample size formulae do not currently exist for when developing a multistate clinical prediction model. Given the combinatorial issues with multistate models, overfitting is of particular concern as the number of individuals passing through some transitions may be small. Future work in this area is therefore paramount. A multistate model, at its core, is a network of cause-specific hazards models [2], which are no different to a

normal time-to-event model. We hypothesise that existing sample size formula could be applied to each model in isolation in order to get a minimum sample size per transition, which could then be divided by the proportion of individuals expected to reach the starting state for that transition in order to derive the total number of individuals required to satisfy that transitions target sample size. The maximum across all transitions would then be taken. For clock-forward models, this may be complicated by the fact that each cause-specific model is an interval censored model, and it is currently unclear how to apply existing sample size formula [34] to interval censored data.

In summary, *calibmsm* provides tools to assess the calibration of the transition probabilities of a multistate model or competing risks model using three approaches (BLR-IPCW, MLR-IPCW and pseudo-values). Further comparison of these approaches in targeted simulations to establish their performance under different censoring mechanisms and assumptions would be valuable. Future work will aim to develop methodology for other model evaluation metrics and incorporate these into *calibmsm*.

## Supporting information

**S1 File. R: a replication script for the analyses.**
(R)

## Author contributions

**Conceptualization:** Alexander Pate, Glen P. Martin.

**Formal analysis:** Alexander Pate.

**Funding acquisition:** Glen P. Martin.

**Methodology:** Alexander Pate, Matthew Sperrin, Richard D. Riley, Ben Van Calster, Glen P. Martin.

**Software:** Alexander Pate.

**Supervision:** Glen P. Martin.

**Writing – original draft:** Alexander Pate.

**Writing – review & editing:** Matthew Sperrin, Richard D. Riley, Ben Van Calster, Glen P. Martin.

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
