## [Decision Letter · Decision Letter 0]

29 Sep 2024

PONE-D-24-31812calibmsm: An R package for calibration plots of the transition 4 probabilities in a multistate modelPLOS ONE

Dear Dr. Pate,

Thank you for submitting your manuscript to PLOS ONE. After careful consideration, we feel that it has merit but does not fully meet PLOS ONE’s publication criteria as it currently stands. Therefore, we invite you to submit a revised version of the manuscript that addresses the points raised during the review process.

Authors need to address all points raised by the reviewers as attached. Additionally, the following points need to address.

**Major Points:**

**Clarity and Justification of Calibration Methods:** The methods section is comprehensive but could benefit from additional clarity regarding the rationale behind choosing the three calibration methods (BLR-IPCW, MLR-IPCW, and pseudo-values). A more detailed comparison or discussion of their relative merits, particularly for different types of datasets (e.g., varying amounts of censoring), would add value.**Handling of Censoring:** Censoring is a significant challenge in survival analysis. The manuscript describes using inverse probability of censoring weights (IPCW) to handle informative censoring, which is a valid approach. However, further discussion on how the choice of weights impacts the calibration curves and the results should be provided. Some sensitivity analysis regarding the impact of weight selection would enhance the robustness of the findings.**Sample Size and Calibration:** The manuscript notes that calibration results could be improved with a larger sample size, but it would be helpful to provide more concrete guidance or examples of how sample size affects calibration estimates. Including a brief power calculation or similar quantitative justification for sample size adequacy would enhance the practical use of the software.**Practical Interpretation:** While the results section demonstrates the use of the **calibmsm** package well, a deeper interpretation of the results (e.g., how clinicians or statisticians should act based on poor calibration) would be helpful. This would provide the necessary bridge between technical calibration results and clinical decision-making.

**Minor Points:**

**Software Documentation:** It would be helpful to include more detailed instructions or a user manual within the manuscript or supplementary materials, especially for users less familiar with R.**Notations and Definitions:** Ensure consistent use of notation, particularly in sections where different methods are discussed. Some readers may find the jump between calibration techniques disorienting without a clear distinction between them.**Figures and Plots:** The calibration curves and scatter plots (such as those generated using BLR-IPCW and MLR-IPCW) should be labeled more clearly in the figures. This will help non-expert readers follow the results more easily.

We look forward to receiving your revised manuscript.

Kind regards,

Md Hasinur Rahaman Khan, Ph.D.

Academic Editor

PLOS ONE

“This work was supported by funding from the MRC-NIHR Methodology Research Programme [grant number: MR/T025085/1].”

Additional Editor Comments:

Authors need to address the points as raised by the two reviewers and given in the attachments. Additionally, I have the following comments that also need to address.

Major Points:

1. Clarity and Justification of Calibration Methods: The methods section is comprehensive but could benefit from additional clarity regarding the rationale behind choosing the three calibration methods (BLR-IPCW, MLR-IPCW, and pseudo-values). A more detailed comparison or discussion of their relative merits, particularly for different types of datasets (e.g., varying amounts of censoring), would add value.

2. Handling of Censoring: Censoring is a significant challenge in survival analysis. The manuscript describes using inverse probability of censoring weights (IPCW) to handle informative censoring, which is a valid approach. However, further discussion on how the choice of weights impacts the calibration curves and the results should be provided. Some sensitivity analysis regarding the impact of weight selection would enhance the robustness of the findings.

3. Sample Size and Calibration: The manuscript notes that calibration results could be improved with a larger sample size, but it would be helpful to provide more concrete guidance or examples of how sample size affects calibration estimates. Including a brief power calculation or similar quantitative justification for sample size adequacy would enhance the practical use of the software.

4. Practical Interpretation: While the results section demonstrates the use of the calibmsm package well, a deeper interpretation of the results (e.g., how clinicians or statisticians should act based on poor calibration) would be helpful. This would provide the necessary bridge between technical calibration results and clinical decision-making.

Minor Points:

1. Software Documentation: It would be helpful to include more detailed instructions or a user manual within the manuscript or supplementary materials, especially for users less familiar with R.

2. Notations and Definitions: Ensure consistent use of notation, particularly in sections where different methods are discussed. Some readers may find the jump between calibration techniques disorienting without a clear distinction between them.

3. Figures and Plots: The calibration curves and scatter plots (such as those generated using BLR-IPCW and MLR-IPCW) should be labeled more clearly in the figures. This will help non-expert readers follow the results more easily.

Reviewers' comments:

Reviewer's Responses to Questions

**Comments to the Author**

1. Is the manuscript technically sound, and do the data support the conclusions?

Reviewer #1: Yes

Reviewer #2: Yes

2. Has the statistical analysis been performed appropriately and rigorously? 

Reviewer #1: Yes

Reviewer #2: Yes

3. Have the authors made all data underlying the findings in their manuscript fully available?

Reviewer #1: Yes

Reviewer #2: Yes

4. Is the manuscript presented in an intelligible fashion and written in standard English?

Reviewer #1: Yes

Reviewer #2: Yes

5. Review Comments to the Author

Reviewer #1: The paper presents an R package, calibmsm, for producing calibration plots for the estimated transition probabilities in multistate regression models. For the most part, the underlying theory relating to the plots has already been published elsewhere, although the current paper extends the approach to any starting time and any starting state (rather than the initial state at time 0) using landmarking. The current paper is structured as a tutorial paper which uses a dataset on leukemia patients following bone marrow transplantation as an illustrative example.

The calibration plots are a very useful diagnostic tool for assessing goodness-of-fit in multistate models with right-censoring. The package works closely with mstate, which is the principal package for fitting such models in R. Generally the paper, and the associated R package, are well written and fairly easy to follow.

Main comments

1. For useability, it might be helpful for the paper to give a bit more detail on how the data frames containing the transition probability estimates are created and the requirements for the data.raw dataset. In the paper, the relevant dataframes are just directly used. However, in practice researchers would need to construct these themselves starting from a fitted mstate model. For the data.raw dataset it is unclear whether the wide form transition times are necessary, or whether it is only the covariates and event time/status for the censoring model.

2. The vignette on IPCW suggests that the default method of calculating the weights is inappropriate. In particular, the "year" variable clearly sets an upper limit on the administrative censoring (and hence overall censoring) time. It would be helpful to see what impact there is if a more appropriate set of weights is used.

Minor comments (page numbers refer to the pages in the generated pdf)

P9 Title page: "Surname" is included in the list of authors.

P17 l194: cox -> Cox

P18 l225: "All multistate models must have an absorbing state": I don't think that is necessarily true. While most applications would include death as an absorbing state, some models (e.g. models of STIs) may have two or more recurrent states and assume that death is negligble or may be treated as non-informative censoring.

P32 l448: Unfinished sentence.

P51 Figure 4: The plots do not appear to have any estimated calibration curves included.

Reviewer #2: The authors present the R-package calibmsm for calibrationsplots of the transition probabilities in a multistate model. This is a quite interesting tool for many researchers that facilitates the application of existing methods. I appreciate the Reference list provided and the overall explanation of the package.

However, I have some minor remarks that might help to improve the quality of manuscript.

- Authors name of Ben van Calster seems to be entered incorrect, as in the authors list “Surname” is written

- Line 30: There is a typo: psuedo -> pseudo

- Line 36: “… the calibration of a multistate model developed…” I think you can say “… any multistate model” in order to underline the flexibility of this package.

- Line 67-68: I think there is a grammatical error in this sentence (“which”?). Please check.

- line 102-103: I would be interested in the transition probabilities into any state. This is actually what you are doing. However, in this sentence it sounds like you are only addressing the transitions out of the starting state.

- Line 228: There is a typo: This is issue…-> delete is

- The numbers referencing to formulas (like (1) and (2)) look the same as the numbers for the references. This is irritating, please use different styles.

- Line 265 – 273: You nicely explain step by step how to estimate confidence intervals. For me as a potential user it would be nice if you could add the information where I can find some example code (maybe as supplement, or in the practical part that follows)

- Section 3 and 4:

o For me it was not that easy to capture the overview provided of section 3 (description of package functions and interface).

o It might help, if you separate it a little bit. Maybe first state everything that is needed. And explain afterwards how it should look like. Maybe consider a step by step approach. In the end there should be a clearer structure in this section 3.

o In general, for a user it is easier to directly see what is goin on (what input for what purpose etc) by including snippets of examples. You could think of combining section 3 and 4. But if you want to keep this separated you could already refer to the next section.

- Figure 4: There seems to be an error when uploading the figure. When viewing the submission I there are only empty graphs (with diagonals).

- Line 548: The authors mention a range of other models, that can be addressed via calibmsm, not only multistate models, e.g. competing risks models. A competing risk model is actually a simple multistate model with 3 states.

6. PLOS authors have the option to publish the peer review history of their article (what does this mean? ). If published, this will include your full peer review and any attached files.

**Do you want your identity to be public for this peer review?** For information about this choice, including consent withdrawal, please see our Privacy Policy .

Reviewer #1: No

Reviewer #2: No

---

## [Author Response · Author response to Decision Letter 1]

8 Nov 2024

Hello, we thank the reviewers and editor for taking the time to read the paper carefully, and giving insightful comments. We have provided a point-by-point response to all comments, our responses in blue text, and changes to the manuscript given in green text, to help the reviewers navigate the document as easily as possible. Line numbers refer to the tracked changes version of the manuscript.

Also, in order for the submission to meet PlosOne submission guidelines we have had to change the document format. Previously, it was generated from within R, as a vignette, however it is required to submit a TEX or .docx file. We have therefore converted the previously submitted pdf to a .docx file, and have applied tracked changes to this. During the conversion it has proved difficult to retain to retain the Sweave of manuscript text, R code and R output. We apologise for any inconsistencies in the formatting (i.e. some of the code + code output looks a bit ugly).

Finally, we have made some changes to the package over the last couple of months. For example, there are new vignettes describing some of the sensitivity analyses we have done as a result of this peer review. The submitted manuscript here reflects the newest version of calibmsm on GitHub, not the version currently on CRAN. We will update the version on CRAN, if/when this manuscript is accepted, to avoid repeatedly updating and pushing to CRAN.

Many thanks,

Editor comments:

Major Points:

Clarity and Justification of Calibration Methods: The methods section is comprehensive but could benefit from additional clarity regarding the rationale behind choosing the three calibration methods (BLR-IPCW, MLR-IPCW, and pseudo-values). A more detailed comparison or discussion of their relative merits, particularly for different types of datasets (e.g., varying amounts of censoring), would add value.

We have added the following text to highlight why these three methods were chosen:

Line 120: These approaches were previously proposed and evaluated in a simulation study,24 but were restricted to assessing calibration out of the starting state at time s=0. The theory is summarised and revised to allow assessment of calibration out of any state j at any time s ofin sections 2.2 – 2.6.

In our previous work we found that both the BLR-IPCW and pseudo-value methods had similar levels of bias and variance under the different censoring mechanisms considered in our simulation,1 therefore they do not have relative merits in this sense. We do know that both methods work when their assumptions are met (conditional independence given some set of baseline covariates Z). We have therefore added the following text to:

1) detail around exactly when these assumptions hold

2) highlight the benefit doing MLR-IPCW as well as either BLR-IPCW or pseudo-value (strength of calibration)

3) highlight the importance of doing both BLR-IPCW or pseudo-value:

Line 794: All three methods (BLR-IPCW, MLR-IPCW and pseudo-value) have been shown to give an unbiased assessment of calibration under random censoring mechanisms, and a predominately unbiased assessment of calibration when there is a strong association between the outcome and censoring mechanisms that can be explained by baseline covariates.1

Line 813: For now, we reiterate the importance of implementing these methods in settings where the observation process/censoring mechanism does not change depending on the outcome state an individual is in. It has previously been suggested to evaluate calibration using MLR-IPCW and one of the BLR-IPCW or pseudo-value approaches because MLR-IPCW provides a stronger assessment of calibration.24 However, we now suggest to evaluate calibration using all three methods, and a comparison between the BLR-IPCW and pseudo-value approaches can be used as a proxy for assessing whether the above assumptions of either method may be violated.

We also direct you towards some existing text around the computational burdens when estimating confidence intervals for each approach:

Line 834: The BLR-IPCW, MLR-IPCW and pseudo-value approaches have different computational burdens. A calibration curve can be obtained reasonably quickly using the BLR-IPCW or MLR-IPCW approaches, however estimation of confidence intervals for BLR-IPCW using bootstrapping (the recommend method in section 2.6) will result in a high computational time in large validation datasets. On the contrary, obtaining the calibration curve itself using the pseudo-value approach has a high computational burden due to estimation of the pseudo-values. Once these have been calculated, a calibration curve and confidence interval can be estimated quickly using parametric techniques, meaning estimation of the confidence interval adds minimal computational burden.

Handling of Censoring: Censoring is a significant challenge in survival analysis. The manuscript describes using inverse probability of censoring weights (IPCW) to handle informative censoring, which is a valid approach. However, further discussion on how the choice of weights impacts the calibration curves and the results should be provided. Some sensitivity analysis regarding the impact of weight selection would enhance the robustness of the findings.

This links to a comment by reviewer 1. As a result, we have undertaken a comprehensive investigation into this and added a large vignette (Sensitivity-analysis-for-IPCWs) detailing our findings.

In summary, firstly, the main analysis were repeated but censoring individuals at 5-years before fitting the model to estimate the weights (i.e. a stopped cox2). This removes the differential cap across the year groups. There is functionality within the package to do this (w_max_follow argument), and is someone we should have done in the first place. This has been detailed in section 4:

Line 438: The w_max_follow=t_eval argument censors individuals at t_eval before fitting the model used to estimate the weights, i.e. a "stopped cox" approach.41 This decision was made to help meet the proportional hazards assumption as there is differential follow up for individuals in different year groups (see vignette Sensitivity-analysis-for-IPCWs27 for more details).

A sensitivity analysis was then done estimating the weights using a flexible parametric model, as opposed to cox proportional hazards, and lead to similar calibration curves. Further investigation then identified the censoring mechanism, and a non-random observation process, to be the probable driving factor behind the difference in the calibration plots for state 1 and state 3. As well as the new vignette, we have detailed this in the methods section:

Line 184: Note that if the censoring mechanism is not conditionally independent from the outcome process X(t) given Z, i.e. the rate of censoring changes depending on outcome state occupancy, then this approach will be invalid. Instead, the outcome history up until time t must be conditioned on when estimating the weights, as specified in equation (3).

Summarised this in the results section:

Line 612: We explored this theory in more detail (see vignette Sensitivity-analysis-for-IPCWs), but found little change when estimating the weights using a flexible parametric survival model. Instead, we identified that this may be caused by a difference in the censoring mechanism for individuals in the adverse event state, as it appeared these individuals were less likely to be censored. This will bias the results from the BLR-IPCW and MLR-IPCW methods unless the weights are conditional on the amount of time spent in each outcome state, something which calibmsm is not currently set up to do. Although it’s not possible to be certain that the individuals in the adverse event state were less likely to be censored purely from looking at the data, we concluded it was a strong possibility, and that the BLR-IPCW calibration curves may be biased in this particular clinical example.

And added the following text to the discussion:

Line 800: This is an indicator that the assumptions underpinning either one of the methods could be violated. This was explored in detail (see vignette Sensitivity-analysis-for-IPCWs27) and led to the conclusion that the BLR-IPCW and MLR-IPCW plots are likely unreliable, in particular for the adverse event state. We hypothesised this was driven by a differential censoring mechanism/observation process for individuals in the adverse event state. Simulations studies are required to 1) quantify this type of bias, and 2) explore whether this can be accounted for by estimating the inverse probability of censoring weights using approaches which are conditional on the time spent in each outcome state (for example a latent-class model). If such a study could be undertaken this would be highly valuable.45,46 For now, we reiterate the importance of implementing these methods in settings where the observation process/censoring mechanism does not change depending on the outcome state an individual is in. It has previously been suggested to evaluate calibration using MLR-IPCW and one of the BLR-IPCW or pseudo-value approaches because MLR-IPCW provides a stronger assessment of calibration.24 However, we now suggest to evaluate calibration using all three methods, and a comparison between the BLR-IPCW and pseudo-value approaches can be used as a proxy for assessing whether the above assumptions of either method may be violated.

Note, we have also removed the BLR-IPCW calibration curves for predictions made at time s = 100, and focus on pseudo-value calibration curves in section 4.3.

Sample Size and Calibration: The manuscript notes that calibration results could be improved with a larger sample size, but it would be helpful to provide more concrete guidance or examples of how sample size affects calibration estimates. Including a brief power calculation or similar quantitative justification for sample size adequacy would enhance the practical use of the software.

This is a pertinent point. Regrettably, there is currently no established way for deriving minimum sample sizes for multistate models. This is an active research topic which a PhD student in our group is working on. We fear that doing a simplistic/brief sample size calculation (I.e. such as 10 events per predictor variable, or some other similar approach), can inadvertently lead to bad practice, if other researchers than read this as being a viable solution in the future. For example, Fishers throwaway comment about p-value of 0.05 becomes standard statistical practice for 50 years! We have therefore avoided an explicit sample size calculation, which we believe would be incorrect, and instead added a deeper discussion about how a sample size calculation could work, and have incorporated this into the future work section:

Line 861: Despite sample size formulae being available for clinical prediction models predicting continuous,5 binary,6,7 time-to-event6 and multinomial outcomes8; sample size formulae do not currently exist for when developing a multistate clinical prediction model. Given the combinatorial issues with multistate models, overfitting is of particular concern as the number of individuals passing through some transitions may be small. Future work in this area is therefore paramount. A multistate model, at its core, is a network of cause-specific hazards models,9 which are no different to a normal time-to-event model. We hypothesise that existing sample size formula could be applied to each model in isolation in order to get a minimum sample size per transition, which could then be divided by the proportion of individuals expected to reach the starting state for that transition in order to derive the total number of individuals required to satisfy that transitions target sample size. The maximum across all transitions would then be taken. For clock-forward models, this may be complicated by the fact that each cause-specific model is an interval censored model, and it is currently unclear how to apply existing sample size formula6 to interval censored data.

Practical Interpretation: While the results section demonstrates the use of the calibmsm package well, a deeper interpretation of the results (e.g., how clinicians or statisticians should act based on poor calibration) would be helpful. This would provide the necessary bridge between technical calibration results and clinical decision-making.

We thank the editor for highlighting this important point. We are pleased that the clinical example shows how to use the package well; as such, we have kept the text around this unchanged, since this is one of the main intended contributions of this paper. However, we fully agree with the reviewer of the need to bridge technical results and clinical decision-making. As such, we have added a detailed discussion of this to a new Box (Box 1) at the end of section 4. This box is intended to outline how researchers should interpret the calibration results across all three approaches, and to give some indication of how researchers should act upon finding that the model is miscalibrated. We emphasize that miscalibration means the model should not be used to inform clinical decision-making, and that revisions to the model would be needed to allow this. We suggest some ways to achieve this (e.g., reducing model complexity to reduce overfitting), but we also note that this itself requires further methodological development.

Specifically, our new Box 1 reads as follows:

Line 776: Box 1: Interpretation of the calibration results.

Assessing the calibration of multi-state clinical prediction models requires consideration of each of the states of the model, with a requirement for there to be good calibration across all states before the model could be used in clinical practice. We have provided three methods to assess calibration (see Section 2), and we recommend assessing calibration using each, so that the results can be compared.

The calibration curves shown in Figure 3, Figure 4 and Figure 5, which consider predictions out of the recovery state a time 0, show that there is good agreement between the observed and predicted risks for some – but not all – of the states. The results tell us that the transition probabilities of remaining in state 1 (transplant) are pre-dominantly over-predicting. Specifically, the model is over-estimating the predicted risk of someone not recovering, having an adverse event, experiencing relapse or death following transplant. The transition probabilities of being in state 2 (recovery) or state 5 (adverse event + recovery) are either under or over-predicting depending on the predicted risk value. A key clinical outcome in this clinical setting is the risk of relapse (state 5), with these results showing that the model should not be used to inform risk estimation for this state, since the calibration of state 5 is extremely poor.

On the contrary, the calibration of transition probabilities for state 4 (adverse event + recovery) and state 6 (death) are reasonably well calibrated. Checking for consistency in conclusion across the three calibration method is always recommended as it may reveal important insights from the analysis. Indeed, we found differences in the calibration results of state 3 across the three methods (as discussed in the main paper). This led to further investigation, which concluded that the calibration plots from the BLR-IPCW and MLR-IPCW approaches may be biased in this setting, in particular state 3 (adverse event), leading us to focus on the pseudo-value calibration plots, which indicates the transition probabilities into the adverse event state were well calibrated.

The pseudo-value calibration curves in Figure 6 and Figure 7, which consider predictions out of the recovery and adverse event states at time 100, show very poor agreement between the observed and predicted risks.

In our opinion, finding that there are some states with miscalibrated transition probabilities informs us that the predicted risks from the model should not be used to inform clinical decision-making. For example, it is clear that the model should not be used to aid clinical decision-making around relapse risk following transplant, especially when making predictions 100 days post-transplant. On the contrary, one ma

---

## [Editor Report · Decision Letter 1]

20 Feb 2025

calibmsm: An R package for calibration plots of the transition probabilities in a multistate model

PONE-D-24-31812R1

Dear Dr. Pate,

We’re pleased to inform you that your manuscript has been judged scientifically suitable for publication and will be formally accepted for publication once it meets all outstanding technical requirements.

Kind regards,

Md Hasinur Rahaman Khan, Ph.D.

Academic Editor

PLOS ONE

Additional Editor Comments (optional):

Need to check carefully the typos and PLOS ONE manuscript structures while providing the final document for potential publication.
---

## [Editor Report · Acceptance letter]

PONE-D-24-31812R1

PLOS ONE

Dear Dr. Pate,

I'm pleased to inform you that your manuscript has been deemed suitable for publication in PLOS ONE. Congratulations! Your manuscript is now being handed over to our production team.

Kind regards,

on behalf of

Dr. Md Hasinur Rahaman Khan

Academic Editor

PLOS ONE